# Past and Future Strategies to Inhibit Membrane Localization of the KRAS Oncogene

**DOI:** 10.3390/ijms222413193

**Published:** 2021-12-07

**Authors:** Malak Haidar, Patrick Jacquemin

**Affiliations:** De Duve Institute, Université Catholique de Louvain, 1200 Brussels, Belgium; malak.haidar@uclouvain.be

**Keywords:** RAS, KRAS, plasma membrane, cancer

## Abstract

KRAS is one of the most studied oncogenes. It is well known that KRAS undergoes post-translational modifications at its C-terminal end. These modifications are essential for its membrane location and activity. Despite significant efforts made in the past three decades to target the mechanisms involved in its membrane localization, no therapies have been approved and taken into the clinic. However, many studies have recently reintroduced interest in the development of KRAS inhibitors, either by directly targeting KRAS or indirectly through the inhibition of critical steps involved in post-translational KRAS modifications. In this review, we summarize the approaches that have been applied over the years to inhibit the membrane localization of KRAS in cancer and propose a new anti-KRAS strategy that could be used in clinic.

## 1. Introduction

The *RAS* oncogenes (*KRAS*, *HRAS*, and *NRAS*) are frequently mutated in human cancers, with *KRAS* being the most commonly affected [1]. RAS are small GTPases that act as binary molecular switches by cycling between an active form bound to guanosine triphosphate (GTP) and an inactive form bound to guanosine diphosphate (GDP)-bound. The switching process and activity of RAS are regulated by guanine nucleotide-exchange factors (GEFs) and GTPase-activating proteins (GAPs) [2]. GEFs induce the release of GDP to GTP from G-proteins by modifying the nucleotide binding site, thereby changing their conformation and increasing their binding affinity for effector proteins [3]. On the other hand, GAPs enhance the hydrolysis of bound GTP to GDP of G-proteins blocking their activities and stopping their downstream signaling pathways [4].

RAS family members are very similar in their catalytic domain, but differ in their hypervariable regions (HVR) (amino acids (aa) 166–185). The catalytic domain (1-86 aa), which has 100% homology between the RAS proteins, while the HVR, which is restricted to the last C-terminal aa, has 10–15% aa identity between RAS proteins, with site-specific amino acid variations that could affect intramolecular dynamics [5]. Oncogenic mutations, particularly at positions 12, 13 or 61, block the negative regulation of RAS by GAPs, and induce its activity. In the RAS family, the *KRAS* gene is the only one that is alternatively spliced at the fourth exon, resulting in two isoform proteins: KRAS4A and KRAS4B. The latter two have identical sequences except for their carboxyl terminus, which required for post-translational modifications and intracellular transport [6]. The respective roles of the two splice isoforms in tumorigenesis are still unclear, with some studies suggesting a preponderant role for KRAS4B [7,8,9].

RAS proteins are located at the inner surface of the plasma membrane. This position is crucial for their biological activity and depends on sequence motifs present in the HVR. The first motif, which is common to all RAS proteins, is the C-terminal CAAX motif (C stands for cysteine, A for aliphatic amino acid, X for any amino acid). This motif undergoes a series of modifications required for the localization to the membrane [10,11]. The initial step requires the addition of 15 carbon farnesyl polyisoprene lipid by farnesyltransfeRASe (FTase) to the cysteine via a stable thioether linkage. As a result, the farnesylated CAAX motif is cleaved off by the farnesylcysteine-directed endoprotease, RAS-converting enzyme (RCE1) that removes the AAX amino acids [12]. Afterwards, the carboxyl group of the newly C-terminal farnesylcysteine undergoes a methylesterification by isoprenylcysteine carboxylmethyltransfeRASe (ICMT) to produce RAS proteins with hydrophobic tails that prevent plasma membrane repulsion [13]. The second motif is either a site present on HRAS, NRAS, and KRAS4A required for palmitoylation by palmitoyl transfeRASe [14], or a polybasic sequence of multiple lysines found on KRAS4B. These lysine residues confer electrostatic interactions with the negatively charged head groups of the plasma membrane lipids [15]. In addition, KRAS4B interacts with phosphodiesteRASe-δ (PDEδ) to promote its translocation to the plasma membrane [16].

The central role that *RAS* genes play in tumorigenesis makes them the targets of choice for cancer therapies. Unfortunately, despite enormous efforts, targeting RAS mutations and developing clinically approved drugs has not been successful, leading some to refer to oncogenic *RAS* mutations as “undruggable”. Since RAS proteins require association with the membrane for their biological activity, many approaches have been developed to disrupt this association and thus block their oncogenic function (Figure 1). Here, we focus on the various methods that have been applied to target KRAS4A and KRAS4B by interfering with their membrane association (Table 1) and trafficking, and highlight recent advances in targeting these processes. These advances lead us to propose here a new strategy to inhibit KRAS membrane localization, raising hopes that KRAS may be “treatable” after all.

## 2. Inhibition of CAAX Prenylation

Since CAAX prenylation is required for oncogenic transformation [17], many efforts have been made to target this process leading to the development of a large number of farnesyltransfeRASe inhibitors (FTIs) drugs, such as L-739,750 [18], FTI-276 [19], or lonafarnib. Many agents entered into clinical, with tipifarnib being the most studied FTI, with phase III clinical trials in cancer [20,21].

Despite promising results in preclinical studies showing that FTIs can effectively suppress tumor growth with low toxicity, the results of clinical trials have not been satisfactory. FTIs were found to lack anti-tumor effect in *KRAS* (and *NRAS*) mutant cancers. This led to the conclusion that blocking KRAS membrane association may be a misapplication. However, a broader understanding of KRAS modification and trafficking has shown that the failure is due to the alternative geranylgeranylation of KRAS by geranylgeranyltransfeRASe (GGTase) when farnesylation is impaired, thereby restoring membrane association [22,23,24,25]. Interestingly, FTIs were found to be effective in cancers with oncogenic H-RAS in the absence of alternative prenylation [22].

The contributions of GGTase-I to RAS prenylation have been validated by both pharmacological and molecular approaches [26]. An inhibitor of GGTase-I (GGTI-2418) was tested in phase I clinical trials but was discontinued due to lack of efficacy in patients [27]. The ability of FTase to bypass inhibition of GGTase and vice versa makes monotherapy with these inhibitors quite challenging. Therefore, it has been investigated whether effective inhibition of prenylation and localization of KRAS is possible by dual inhibition of FTase and GGTase-I. Hence, considerable efforts have been made to synthesize dual prenyltransfeRASe inhibitors. L-778123, for instance, has probably been the most studied. It has entered clinical trials partly because it can completely inhibit KRAS prenylation. Notwithstanding the confirmation of the pharmacological profile of L-778123 in humans as a dual inhibitor of FTase and GGTase-I, L-778123 was to not able to inhibit KRAS prenylation in vivo [28].

Nevertheless, a dual inhibitor (FGTI-2734) has recently been developed and has shown to prevent membrane localization of KRAS, overcoming the previous problem of resistance of KRAS to prenylation inhibition. Using pharmacologic and genetic approaches, it has been shown that the cytosolic KRAS is still able to bind RAF-1 but cannot activate it, possibly due to inhibition of binding to the scaffold protein KSR, which plays an important role in RAS activation of RAF [29]. These results warrant further preclinical and clinical studies with this compound.

In addition, statin was recently shown to enhance the immunogenicity of *KRAS*-mutant cancer by inhibiting its prenylation. In addition, statin administration in combination with oxaliplatin, an immunogenic cell death inducer, was found to elicit an effective antitumor immune response in genetically engineered *KRAS* tumor models. Furthermore, the addition of checkpoint inhibitors to the aforementioned combination therapy sensitizes tumors to PD-1 blockade, triggering tumor suppressive effects [30].

## 3. Targeting the Post-Prenylation CAAX Processing

It is known that post-prenylation CAAX processing enzymes contribute to RAS membrane association, making them therefore attractive targets for drug development. The activity of RCE1 is critical for the proper localization of RAS to the plasma membrane. Studies using in vivo and in vitro reporters of RCE1 activity have shown that enzyme activity depends on the prenylation status of the substrates and the identity of the aliphatic amino residues. The second amino residue (A2) in the CAAX sequence is most critical for recognition, and favors Ile, Leu, or Val [31]. There are several categories of RCE1 inhibitors, namely non-specific protease inhibitors, substrate mimetics, including natural products, and small-molecule inhibitors. 

Non-specific inhibitors include an irreversible serine/cysteine-protease inhibitor, called Nα-tosyl-l-phenylalanine chloromethyl ketone (TPCK) [32,33], and organomercurials (para-hydroxymercuribenzoic acid, para-hydroxymercuriphenylsulfonic acid, and mersalyl acid) [32,34]. Substrate mimetics that inhibit RCE1 are based on isoprenylated peptides, or isoprenoids. RPI was the first non-hydrolyzable isoprenylpeptide to be described; it mimics the CAAX motif of KRAS4B [35]. Later on, many others isoprenylated peptides were developed; however, they were all less potent compared to RPI [36]. In addition, isoprenyl mimetic substrates that possess farnesyl substituent with a polar headgroup have been defined. Barangcadoic acid A and rhopaloic acids, for example, are both marine natural products derived from Hippospongia species that have a farnesyl substituent and can inhibit RCE1 [37]. BFCCMK and UM96001 are two chloromethyl ketones that possess a farnesylated cysteine and can effectively inhibit the growth of RAS-transformed rodent and human cancer cells [38,39]. 

Small-molecule inhibitors are preferred over peptide and isoprenoid-based inhibitors as they are expected to have better cell permeability, stability and easier chemical synthesis. A library screening for farnesyl transferase inhibitors (FTIs) identified non-peptidic and non-prenylic inhibitors of RCE1 termed Non-Peptidic Protease Inhibitor (NPPI-A, B, and C) [40]. Another screening approach identified nine small-molecule inhibitors of RCE1, including NSC 1011, 73101, 295642, 321237, and 609974 [41]. Recently, it has been published that the anti-HIV protease inhibitors Ritonavir and Lopinavir can suppress RCE1 and CAAX Rab proteins, sensitizing the liver to organelle stress and injury [42]. 

None of the above inhibitors have yet produced encouraging results in preclinical tests, raising the question of whether inhibiting RCE1 is an attractive strategy for inhibiting KRAS. At the very least, further studies are needed to better understand exactly how RCE1 functions. While the topology and crystal structure of RCE1 are known from the yeast *Saccharomyces cerevisiae* [43] and the archaea *Methanococcus maripaludi* [44], the structure of human RCE1 has not yet been elucidated. This makes the selective inhibition of RCE1 a difficult problem.

In addition to RCE1, ICMT is also known to be involved in the post-prenylation modifications of RAS and its proper cellular localization. Expression of GFP KRAS fusion proteins in homozygous Icmt knockout embryonic stem cells showed an accumulation of KRAS in the cytoplasm [45,46]. This makes ICMT a potent target for anti-cancer therapy. To date, many small molecule compounds have been developed to inhibit ICMT.

Screening of a chemical compound library of about 10,000 compounds led to the identification of a group of compounds with an indole core structure [47]. Cysmethynil is the most potent of these. It is highly specific, as it has not been shown to not inhibit other enzymes involved in the C-terminal modification such as FTase, GGTase-I, or RCE1. The role of cysmethynil in affecting tumor growth is well established in vivo and in vitro [48,49]. In addition, another class of ICMT inhibitors was developed by using a substrate-based approach, which led to the identification of the adamantyl derivative and its analogs [50]. In a complementary approach, FTPA triazole was identified as another potent inhibitor of ICMT1 [51]. 

ICMT inhibitors have been shown to regulate the activation of RAS, making them a potential therapeutic approach against cancer. However, none of the ICMT inhibitors developed to date have been shown to be therapeutically effective in clinical trials. Moreover, inactivation of Icmt enhanced KRAS-induced pancreatic cancer in mice [52], suggesting that inhibition of ICMT activity may be counterproductive in some cancers. Therefore, more studies are still needed to develop a successful therapeutic strategy with ICMT inhibitors.

## 4. Disrupting KRAS4A Palmitoylation

Palmitoylation acts as a second signal to stabilize the membrane association of NRAS, HRAS and KRAS4A. Therefore, its interference could disrupt the activity of these RAS isoforms. Palmitoylation is a post-translational protein modification in which a palmitic acid is attached to cysteine residues via a thioester bond. Protein acyltransferases (PATs), also known as palmitoyltransferases, catalyze this reaction by transferring the palmitoyl group of palmitoyl-CoA to the thiol group of cysteine residues. To the best of our knowledge, there are still no drugs that specifically target palmitoylation, as the development of specific inhibitors has been limited due to the lack of suitable tools.

However, there are some possibilities that deserve to be explored in more detail. Although it is unlikely that 2-bromopalmitic acid (2-BP), a non-metabolizable palmitate used to inhibit palmitoylation, will be developed into a drug given the large number of palmitoylated proteins in humans, efforts continue to be made to develop pharmacological modulators of palmitoylation for the treatment of diseases. This has been especially true since the identification of 23 DHHC (aspartic acid-histidine-histidine-cysteine tetrapeptide motif) proteins in the repertoire of mammalian protein acyltransferases (PATs) [53]. DHHC, initially known as an enzyme with a zinc finger motif, was first identified in yeast in 2002, where it was shown to catalyze the S-palmitoylation of a yeast homologue RAS [54].

Library screening identified more selective inhibitors of palmitoylation, leading to the discovery of five compounds (I–V) that inhibited cellular processes mediated by palmitoylation [55]. However, follow-up studies revealed that only compound V was able to inhibit the activity of all four DHHC proteins tested [56] and PAT auto-acylation [57]. Despite the antitumor activity of the compound V and its ability to inhibit DHHC PATs, clinical trials have not yet been conducted.

Interestingly, a recent work has shown that the palmitoylation status of individual proteins can be selectively altered by manipulating the recruitment of specific substrates to specific PATs [58]. This provides a valuable tool for future studies to profile protein palmitoylation, particularly to explore the possibility of identifying novel inhibitors of KRAS4A.

Since it is known that some palmitoylated proteins undergo enzymatic de-acylation catalyzed by acyl protein thioesterase (APTs) and that two of the APTs (APT1 and APT2) possess depalmitoylating activity [59,60,61,62], further progress has been made in the development of molecules that inhibit depalmitoylation. Palmostatin B (APT1) was the first compound shown to inhibit RAS depalmitoylation in cells [63]. This inhibitor disrupts the cellular acylation cycle at the level of depalmitoylation, causing a loss of the precise steady-state localization of palmitoylated RAS. Consequently, it partially reverses the oncogenic phenotype of RAS-transformed fibroblasts. 

## 5. Targeting KRAS Localization by Perturbing Electrostatic Interaction

Many molecules have been identified that target KRAS membrane localization by modulating the electrostatic interaction between the polybasic domain of KRAS4B and the plasma membrane. The amphiphilic drug chlorpromazine (CPZ) reduces the association of KRAS with the plasma membrane and results in delocalization to the cytoplasmic pools. These effects appear to be dependent on electrostatic interactions generally arising from polybasic domains, as the membrane association of another related protein possessing a membrane-interacting polybasic cluster was also disrupted, but not that of HRAS [64].

Moreover, Staurosporine (STS) has been shown to translocate the oncogenic mutant KRAS from the plasma membrane and abrogate the proliferation of KRAS-transformed cells. The mechanism of action of STS involves disrupting the subcellular localization of phosphatidylserine (PS) by blocking its endosomal recycling. Therefore, it leads to decreased electrostatic potential of the plasma membrane and a concomitant redistribution of KRAS to the early and late endosomes, lysosomes, mitochondria, Golgi apparatus, and endoplasmic reticulum [65].

Fendiline, an L-type calcium channel blocker, was also identified as a specific inhibitor of plasma membrane of KRAS with no detectable effects on the localization of the other RAS isoforms (HRAS and NRAS) [66]. However, the mislocalization of KRAS is calcium-independent, as other classes of L-type calcium channel blockers did not cause mislocalization of KRAS. Fendiline reduced KRAS nanoclustering on the plasma membrane and redistributed KRAS from the plasma membrane to the endoplasmic reticulum, Golgi apparatus, endosomes, and cytosol. Like CPZ, it did not inhibit KRAS posttranslational processing. Rather it perturbs the electrostatic interactions of polybasic domains with the electronegative inner leaflet of the plasma membrane, and consequently impairs the transport of prenylated polybasic domain-targeted RAS proteins. Fendiline significantly abrogates signaling transduction downstream of the constitutively active KRAS and subsequently blocks proliferation of pancreatic, colon, lung, and endometrial cancer cell lines expressing an oncogenic mutant *KRAS* [66].

## 6. Targeting KRAS Interaction with Proteins Required for Its Membrane Localization

Efforts to develop drugs targeting the post-translational modifications of KRAS that regulate its membrane association have continued. An important advance was recently made by targeting the prenyl-binding protein PDEδ, which is required for proper localization and signaling of farnesylated RAS, but not that of KRAS4A.

Suppression of PDEδ levels was found to disrupt the association of RAS with the plasma membrane [67] and to impair the growth of RAS-mutated cancer cells [68]. A high-throughput screen allowed the identification and characterization of a small-molecule inhibitor called deltarasin that blocks PDEδ association with the farnesylated tail of KRAS4B [69]. The ability of deltarasin to block the PDEδ–KRAS4B interaction has been validated in vitro and *in vivo*. Inhibition of PDEδ by deltarasin in human KRAS-mutant pancreatic cancer cell lines blocks the localization of KRAS to the plasma membrane and impairs their proliferative capacity [69]. Moreover, phosphorylation of the ERK1 and ERK2 proteins was found to be significantly reduced after suppression of PDEδ [69]. In addition, treatment with deltarasin reduces tumor growth in a mouse model of pancreatic ductal adenocarcinoma in a dose-dependent manner [69].

Despite the efficacy of deltarasin in regulating the association of KRAS with the plasma membrane, it could have unanticipated consequences. PDEδ may interact with other farnesylated proteins, including farnesylated proteins of the RAS family that act as tumor suppressors. Indeed, supression of such proteins may lead to toxic effects in normal cells. Moreover, PDEδ is also required for farnesylated and geranylgeranylated proteins [70], which may enhance its off-target effects. Moreover, RAS proteins are only partially dependent on PDEδ, as they can still bind to the cell membranes despite its absence [68]. Furthermore, deficiency of KRAS is lethal, not PDEδ [71], confirming that KRAS functions are not completely dependent on PDEδ. Thus, although inhibition of PDEδ appears to be important and promising for the regulation of RAS membranes, much remains to be learned about the function of PDEδ and its ability to block membrane localization of of the RAS mutant.

Another molecule, a specific peptide inhibitor called Memrasin, also blocked the association of KRAS4B to the plasma membrane [72]. Memrasin consists of a membrane -bound sequence derived from the C-terminal region of KRAS4B and an endosome escape motif. It abrogates the binding of KRAS4B to the plasma membrane and impairs RAS signaling activity. Moreover, it has been shown to efficiently reduce the viability of several human lung cancer cells in a dose-dependent and KRAS-dependent manner. Memrasin is a useful tool for exploring the biological function of KRAS4B at or outside the plasma membrane and a potential starting point for further development of anti-RAS therapeutics [72].

In addition to identifying molecules that can inhibit KRAS4B signaling, screening of bioactive lipid libraries using a RAS-specific cell viability assay enables the discovery of a new class of inhibitors of RAS transformation. The identified compound, the endocannabinoid N-arachidonoyl dopamine (NADA), can induce cell oncosis; independently of cannabinoid receptors. It inhibits oncogenic transformation induced by NRAS and KRAS4A by suppressing their plasma membrane translocation. Interestingly, it cannot abrogate that of KRAS4B. NADA appears to act in a palmitoylation-dependent manner [73].

As far as we know, despite these promising properties, no molecule targeting the membrane localization of KRAS in cancer is currently in clinical trial.

## 7. Targeting Downstream Mediators of KRAS Signaling

Since attempts to inhibit KRAS have been unsuccessful in the clinic, targeting key effector cascades of KRAS oncoprotein, particularly the mitogenic RAF-MEK-ERK pathway, represents another interesting strategy. KRAS activation induces RAF protein phosphorylation and dimerization with subsequent activation of their downstream kinases. The RAF family consists of three isoforms: ARAF, BRAF and CRAF [74,75]. Inhibitors targeting BRAF, MEK, or ERK have been developed to block KRAS signaling in cancer. For example, BRAF inhibitors have been shown to be significantly effective in improving overall survival (OS) in patients with BRAF-mutated melanoma [76] and have recently been approved for the treatment of BRAF-mutated non-small cell lung cancer (NSCLC) [77]. However, their therapeutic effect has been limited in KRAS-driven tumors due to the development of drug resistance. BRAF inhibitors bind to BRAF and paradoxically induce transactivation of CRAF with subsequent activation of MEK/ERK signaling [78]. To overcome this resistance, several panRAF inhibitors have been developed [79,80,81]. However, despite the promising effect observed in preclinical studies [82,83], monotherapy with panRAF inhibitors has not shown efficacy in early clinical trials [84]. Nevertheless, experiments are currently underway to evaluate the effects of new panRAF inhibitors, such as LXH254 and belvarafenib, administrated alone or in combination with other drugs, on *KRAS*-driven cancers. LXH254 is being therapeutically evaluated in patients with melanoma; both as monotherapy and in combination with anti-PD-1 (NCT02607813). The role of belvarafenib has been tested in patients with advanced solid tumors with RAS or BRAF mutations and has shown a good safety profile and antitumor activity [85,86]. Belvarafenib continues to be studied in combination with the MEK inhibitor cobimetinib (Clinical trial information: NCT02405065, NCT03118817).

Several MEK inhibitors have been tested in *KRAS*-mutated cancers and have not shown any clinical benefits. For example, selumetinib given alone or in combination with chemotherapy failed to improve progression-free survival (PFS) in patients with advanced *KRAS*-mutant NSCLC [87]. The same results were observed with other MEK inhibitors in pancreatic cancer. A phase II trial evaluated the overall survival of pancreatic cancer patients treated with trametinib (MEK inhibitor) and gemcitabine [88]. Safety end-points such as PFS, overall response rate (ORR) and duration of response (DOR) were also evaluated. The results showed that the addition of trametinib to gemcitabine did not improve OS, PFS, ORR or DOR in patients with previously untreated metastatic pancreatic cancer [88]. 

The selective MEK1/2 inhibitor pimasertib has shown antitumor activity in a pancreatic tumor model. A two-part phase I/II, trial was conducted in patients with metastatic pancreatic adenocarcinoma (mPaCa) (NCT01016483). Patients were treated with pimasertib and gemcitabine and, PFS OS, and safety were assessed. No clinical benefit of first-line treatment with pimasertib plus gemcitabine compared with gemcitabine alone was observed in patients with mPaCa. This lack of efficacy is mainly due to alternative compensatory mechanisms involving reactivation of ERK signaling [89,90]. In addition, many studies have shown that treatment with MEK inhibitor leads to an enhancement of the PI3K/AKT pathway, which is due to hyperactivation of ERBB3 [91,92].

In addition, many ERK inhibitors have been used in clinical trials to treat a variety of advanced solid tumors with RAS, RAF or MAPK pathway alterations [93] (ClinicalTrials.gov Identifier: NCT03415126, NCT02711345). For example, ulixertinib/BVD523 and LY3214996 have been tested in phase I clinical trials (NCT02608229, NCT02857270) and showed promising antitumor activity. Thus, ulixertinib has been shown to effectively inhibit in vitro growth of several pancreatic ductal adenocarcinoma cell lines and potentiate the cytotoxic effect of gemcitabine [94]. Recently, a phase II clinical trial tested the efficacy and safety of BVD-523 in patients with metastatic uveal melanoma [95].

Several studies have shown that cancer cells treated with RAF or MEK inhibitors employ multiple mechanisms to reactivate ERK signaling. Therefore, combinations of ERK inhibitors with MEK or RAF inhibitors have been closely studied and evaluated [96,97]. In addition to activating MAPK signaling, KRAS is also known to activate the PI3K-AKT-mTORC1 pathway, contributing to cancer progression. Therefore, many AKT inhibitors have been developed to inhibit PI3K signaling but failed in in vitro and in vivo trials. However, in combination with MEK, ERK or RAF inhibitors, promising results were observed in early studies, but at the cost of higher toxicity [98,99]. The clinical utility of inhibitors of the RAS-regulated RAF-MEK1/2-ERK1/2 pathway and the AKT pathway as single agents has been demonstrated in KRAS-mutant tumors, so novel strategies of dual or triple inhibitors, may be required. 

In addition to its role in activating MAPK and AKT signaling, KRAS is known to mediate metabolic rewiring in cancer, and many studies have elucidated the mechanisms by which KRAS reprograms cellular metabolism to support tumorigenesis [100]. For example, KRAS was found to promote glutamine metabolism and control pancreatic cancer chemoresistance by upregulating the expression of the antioxidant NRF2 [101]. Furthermore, loss of LKB1 in *KRAS*-mutant lung adenocarcinomas activates KEAP1 and leads to metabolic changes that maintain redox homeostasis and promote glutamine metabolism [102]. KRAS can also induce fatty acid synthase (FASN) to promote lipogenesis, and inhibition of FASN was shown to block cellular proliferation of *KRAS*-mutant lung cancer cells [103]. In addition, *KRAS*-mutated cancer cells have been shown to rely on serum lipids to maintain their proliferation and survival [104]. Overall, KRAS-mediated metabolic reprogramming in cancer offers new therapeutic approaches for the treatment of KRAS-driven cancers.

## 8. Modification of KRAS by Ubiquitination

KRAS subcellular localization has also been shown to be controlled by ubiquitination. The loss of Lztr1 abolishes RAS ubiquitination at lysine-170 leading to inhibition of RAS signaling by reducing its association with the membrane [105]. This indicate that ubiquitination may represent a new therapeutic approach for the treatment of *KRAS*-mutant cancer.

## 9. Conclusions and Perspectives

Progress has been made in the discovery of compounds that block membrane binding of KRAS. Many KRAS inhibitory compounds have been identified, a considerable number of which have been shown to be effective in vitro and in vivo. However, none of them have successfully passed the clinical trial stage. This is likely due to the fact that these compounds target not only KRAS, but also a variety of other factors, causing significant toxicity. This is particularly true for the processing of the C-terminal end of KRAS: the CAAX sequence required for this processing is present in more than 100 other proteins, including tumor suppressors. Inhibition of this processing may therefore in accelerating, rather than slowing, cancer progression, as has been observed in pancreatic cancer [52].

In this context, it is now important to reduce this toxicity. We believe that this requires both the identification of new more selective compounds and the implication of a treatment strategy that combined compounds that have been already identified. This strategy must involve vertical inhibition of multiple steps of the cascade that brings KRAS from the endoplasmic reticulum to the plasma membrane, and target not only KRAS4B, but also KRAS4A, as a recent study has shown a role for this previously neglected splice isoform in lung cancer [106]. This combined vertical inhibition is expected to elicit a strong synergy that will allow effective treatment at lower doses, as has been demonstrated for other cascades [107], overcoming the limitations seen with single-compound treatment. In this context, it will certainly be beneficial to use a cell line that allows simultaneous visualization of the membrane localization of KRAS4A and KRAS4B expressed from their endogenous locus. Such a line can now be derived from KRAS^citrine-G12D^ mice that we have recently generated [108]. Crucial breakthroughs in these directions are now critical to overturn the notion that KRAS is undruggable and ultimately revealing its druggability.

## Figures and Tables

**Figure 1 ijms-22-13193-f001:**
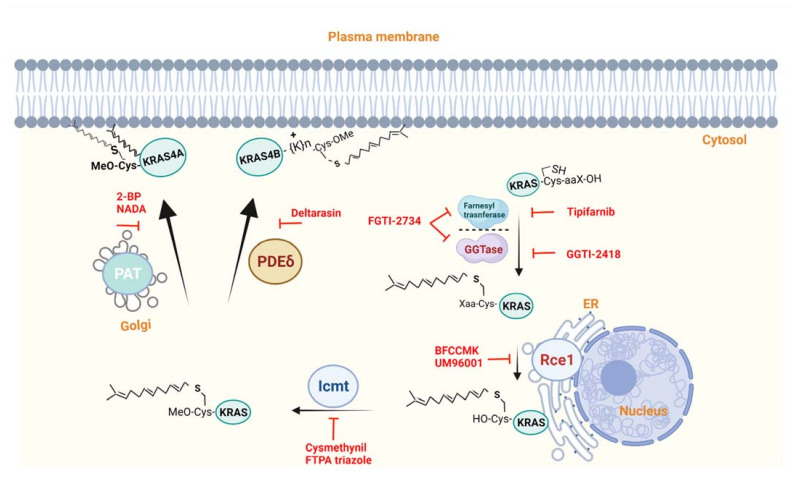
Overview of the post-translational modifications allowing membrane targeting of KRAS4A and KRAS4B and the enzymes involved in these modifications. Effective inhibitors of these enzymes are shown in red. For more details, see the text.

**Table 1 ijms-22-13193-t001:** List of the compounds that inhibit KRAS interaction with the plasma membrane.

Drug	Target Mechanism
Deltarasin	blocks interaction of PDEδ with KRAS4B
Memrasin	direct inhibitor of KRAS4B-PM interaction by forming peptide-enriched domains in the membrane liquid-disordered (ld) microdomains
Fendiline	L-type calcium channel blocker that inhibits KRAS localization to the plasma membrane
FTIs	blocks KRAS membrane association by preventing the addition of prenyl group
staurosporin	inhibits KRAS plasma membrane binding by blocking endosomal recycling of phosphatidylserine
AMG 510	locks KRAS^G12C^ in GDP inactive bound form by binding to its cysteine residue
RCE1 and ICMT inhibitors	blocks post-prenylation processing of KRAS and its membrane association
Statin	inhibits KRAS membrane association via blockage of prenylation
NADA	inhibits KRAS plasma membrane translocation in a palmitoylation dependent manner

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
