# Peer review of "Past and Future Strategies to Inhibit Membrane Localization of the KRAS Oncogene"

_ijms, 2021, doi:10.3390/ijms222413193_

Round 1

Reviewer 1 Report

Haidar and Jacquemin offer an important view on the many efforts targeting the KRAS oncogene, focusing on the strategies impeding KRAS membrane association. The study is well written, appropriately referenced and fully merits publication, however recent findings describing that additional post-translational modifications of RAS with ubiquitin regulate its membrane recruitment and activity are not mentioned. These studies indicate that the ubiquitin system could be also targeted and could be discussed as a future perspective.

Author Response

We thank the reviewers for their complimentary comments on our work. We have completely addressed the points raised and modified the manuscript accordingly.

Please find below our answers to the reviewers’ comments:

A small paragraph has been added to explain the implication of ubiquitin in the regulation of KRAS membrane recruitment and activity. Please see paragraph 8 lines 363 to 368.

Reviewer 2 Report

A timely review article by Dr. Jacquemin and a colleague elaborate on the mechanism of membrane localization of KRAS oncoprotein and ways to utilize this for therapeutic purposes. Few things need to be addressed before it is ready for acceptance. They are as follows:

  1. It has been recently discussed in (PMCID: PMC8045781) how KRAS regulates cancer metabolism and how it can be utilized for therapeutic purposes. In this review under "Targeting Downstream Mediators of KRAS Signaling," a few lines should be added mentioning how targeting cancer metabolism in KRAS-driven cancers would be beneficial for treatment purposes. Since it has been discussed in the above-mentioned article so adding little would be sufficient to meet the need, mentioning it has been discussed elsewhere. But since targeting metabolism is one of the strategies for therapies it will be worthwhile to mention this.
  2. Authors must add a figure depicting the discussed pathways/ strategies to target KRAS-driven cancers by targeting membrane localization. 
  3. The authors should mention if any clinical trial going on by targeting membrane localization in KRAS-driven cancers.
  4. A table consisting summary of the compounds that inhibit K-Ras interaction with the plasma membrane is needed in this review.
  5. It has been shown recently that Statin-mediated inhibition of KRAS prenylation provoked severe endoplasmic reticulum stress by attenuating the anti-ER stress effect of KRAS mutation, thereby resulting in the immunogenic cell death of KRAS mutant cancer cells (PMCID: PMC8327837). This is an interesting observation. The authors should add a few lines discussing this topic. t will be a significant addition to this review. 

Author Response

We thank the reviewers for their complimentary comments on our work. We have completely addressed the points raised and modified the manuscript accordingly.Please find below our answers to the reviewers’ comments:

Reviewer#2:

  1. It has been recently discussed in (PMCID: PMC8045781) how KRAS regulates cancer metabolism and how it can be utilized for therapeutic purposes. In this review under "Targeting Downstream Mediators of KRAS Signaling," a few lines should be added mentioning how targeting cancer metabolism in KRAS-driven cancers would be beneficial for treatment purposes. Since it has been discussed in the above-mentioned article so adding little would be sufficient to meet the need, mentioning it has been discussed elsewhere. But since targeting metabolism is one of the strategies for therapies it will be worthwhile to mention this.

A few extra sentences to cover this notion have been added to the section “Targeting Downstream Mediators of KRAS Signaling” as suggested. Please see lines (347-359).

  1. Authors must add a figure depicting the discussed pathways/ strategies to target KRAS-driven cancers by targeting membrane localization. 

Figure 1 has been added as proposed.

  1. The authors should mention if any clinical trial going on by targeting membrane localization in KRAS-driven cancers.

Few sentences were added to address this comment. Please see lines 281-282

  1. A table consisting summary of the compounds that inhibit K-Ras interaction with the plasma membrane is needed in this review.

A table summarizing the main inhibitors used to inhibit K-Ras membrane targeting has been added now.

It has been shown recently that Statin-mediated inhibition of KRAS prenylation provoked severe endoplasmic reticulum stress by attenuating the anti-ER stress effect of KRAS mutation, thereby resulting in the immunogenic cell death of KRAS mutant cancer cells (PMCID: PMC8327837). This is an interesting observation. The authors should add a few lines discussing this topic. it will be a significant addition to this review. 

A paragraph explaining this topic has been added, please see lines 103-108.

Round 2

Reviewer 2 Report

All concerns have been addressed, ready for acceptance.

This manuscript is a resubmission of an earlier submission. The following is a list of the peer review reports and author responses from that submission.